# Potential Neurocognitive Symptoms Due to Respiratory Syncytial Virus Infection

**DOI:** 10.3390/pathogens11010047

**Published:** 2021-12-31

**Authors:** Catalina A. Andrade, Alexis M. Kalergis, Karen Bohmwald

**Affiliations:** 1Department of Molecular and Microbiology, Faculty of Biological Science, Millennium Institute on Immunology and Immunotherapy, Pontificia Universidad Católica de Chile, Santiago 8331010, Chile; cnandrade@uc.cl; 2Departamento de Endocrinología, Facultad de Medicina, Pontificia Universidad Católica de Chile, Santiago 8331010, Chile

**Keywords:** human respiratory syncytial virus, maternal immune activation, central nervous system, neurological alterations

## Abstract

Respiratory infections are among the major public health burdens, especially during winter. Along these lines, the human respiratory syncytial virus (hRSV) is the principal viral agent causing acute lower respiratory tract infections leading to hospitalization. The pulmonary manifestations due to hRSV infection are bronchiolitis and pneumonia, where the population most affected are infants and the elderly. However, recent evidence suggests that hRSV infection can impact the mother and fetus during pregnancy. Studies have indicated that hRSV can infect different cell types from the placenta and even cross the placenta barrier and infect the fetus. In addition, it is known that infections during the gestational period can lead to severe consequences for the development of the fetus due not only to a direct viral infection but also because of maternal immune activation (MIA). Furthermore, it has been described that the development of the central nervous system (CNS) of the fetus can be affected by the inflammatory environment of the uterus caused by viral infections. Increasing evidence supports the notion that hRSV could invade the CNS and infect nervous cells, such as microglia, neurons, and astrocytes, promoting neuroinflammation. Moreover, it has been described that the hRSV infection can provoke neurological manifestations, including cognitive impairment and behavioral alterations. Here, we will review the potential effect of hRSV in brain development and the potential long-term neurological sequelae.

## 1. Introduction

Globally, respiratory infections are a major public health burden due to the high hospitalizations, morbidity, and mortality rates [1,2]. One of the viruses responsible for causing acute lower respiratory tract infections (ALRTI) is the human respiratory syncytial virus (hRSV) [1,3]. This virus belongs to the *Pneumoviridae* family and contains a negative-sense, single-strain enveloped RNA molecule [4,5]. The respiratory pathologies associated with hRSV infection can vary from rhinorrhea, cough, congestion, low-grade fever, and respiratory distress, leading to more severe symptoms such as bronchiolitis and pneumonia [1,3,5]. The populations affected more severely are infants and the elderly, and the positive cases in infants can reach up to 33 million per year worldwide [6,7]. Virtually all infants under two years old have been infected at least once by hRSV, and the rate of reinfection in these infants is nearly 40% [6,8]. Annually, hRSV infection causes more than three million hospitalizations in children under five years old. Children that are more likely to develop complications during hospitalization are the group possessing high-risk factors, such as preterm birth, chronic lung pathologies, and immunosuppression [2,5]. Since the identification of hRSV, it was established that this virus circulates during the fall-winter period, but lately, some reports have indicated a circulation delay beginning in spring and lasting until summer [9,10,11,12]. These changes in the circulations have been associated with an increase of positive cases [13], which could lead to augmented hospitalization rates. Even though there have been numerous efforts to control infections by this virus, there is no vaccine approved for human use, and the only specific prophylaxis used against hRSV infection is the monoclonal antibody Palivizumab [14,15].

Although the primary targets of the infection with hRSV are the epithelial lung cells, it has been demonstrated that hRSV can also infect cells in the central nervous system (CNS) and the placenta [16,17,18]. Several reports suggest that the infection with hRSV can cause neurological symptoms, such as encephalitis and encephalopathies, leading to long-term neurological consequences [19,20,21]. In addition, some reports evidence that the infection with hRSV can severely affect pregnant women, leading to more severe symptoms during pregnancy [22,23]. Even more, it has been suggested that through the placenta, hRSV can infect the fetus and cause them to develop long-term neurological sequels [24]. This article will discuss the effect of the hRSV-infection during pregnancy, the possible alterations in the development of the CNS of the fetus, and its neurological sequels on the infant.

## 2. hRSV Infection during Pregnancy and Effects on the Newborn

As it is known, the highest risk age group susceptible to viral infection are infants, the elderly, and immunocompromised people [25,26,27,28]. However, during pregnancy, the women increase their probability of having complications from a respiratory viral infection, such as preeclampsia, acute cardiopulmonary diseases, respiratory distress, and pneumonia, among others [29,30]. One of the most studied viruses that impact the health of pregnant women is the Influenza virus [31,32]. This virus affects mainly infants, the elderly, and the immunocompromised, causing symptoms related to the lower respiratory tract -such as cough, sore throat, and pneumonia [33]. Influenza virus can cause severe disease in adults, and as a consequence of this, it is the first suspect of a viral agent in respiratory complications during pregnancy [34]. Contrary to this, little is known about the effect of hRSV infection in pregnant women. This section will discuss the current evidence regarding the infection with hRSV during pregnancy.

The physiological changes in pregnant women, such as cardiopulmonary function and immunological response, explain the increased susceptibility to respiratory viral infection [23]. A study reported three cases of pregnant women who were positive for hRSV, where two of them required intubation and ICU level care [23]. The hRSV infection in these patients exacerbated the pre-existing conditions such as co-infections with other viruses and asthma, which has been previously reported that is susceptible to appear after an episode of bronchiolitis in infants caused by the infection with hRSV [23,35]. As observed in infants, the hRSV infection symptomatology during pregnancy is primarily cough, congestion, and sore throat [36,37]. The incidence of hRSV infection in pregnant women also varies annually during the winter season [36,38,39].

Additionally, an association between the hRSV infection during pregnancy and preterm birth has been reported [22]. Concerning the effects of the hRSV infection during pregnancy, the presence of hRSV has been detected at the birth of a newborn with respiratory distress symptoms [40]. In this case, the patient presented severe chest retraction and poor oxygen saturation, and one week later, the serological test showed the presence of antibodies against hRSV [40]. Moreover, hRSV RNA was found in blood samples from the patient obtained on the first day of life [40]. Interestingly, the serological analysis performed on the mother of the patient showed an elevated titer of antibodies against hRSV, suggesting the possibility of vertical transmission of the virus [40]. Accordingly, a study performed in neonates at birth in which mothers who had a history of respiratory illness during the pregnancy showed anti-hRSV IgA, IgM, and IgG titers in cord blood samples [41]. In these cases, half of the neonates showed respiratory problems that included apnea, respiratory failure, and pneumonia, where 56% of them needed supplemental oxygen [41].

Additionally, other important parameters such as white blood cells and C-reactive protein in serum were high in the neonates with anti-hRSV antibodies compared to controls [41]. These observations led to studies using the BeWoo cell line (human choriocarcinoma cells), where the peak of the viral load detected was at 24 hrs post-infection applying MOI 5 [18]. Moreover, the hRSV infection of BeWoo cells produce infective particle and express the known receptor for the virus, TLR4, and nucleolin [18]. Furthermore, the hRSV infection was evaluated in three cell types (cytotrophoblast, fibroblasts, and Hofbauer cells) obtained from human placental villous tissue [42]. By using a recombinant hRSV expressing the red fluorescent protein (RFP) upon replication (rrhRSV) was described that the virus infects the fibroblasts and Hofbauer cells, but not the cytotrophoblast (Figure 1) [42]. Besides, the kinetics of the infection on Hofbauer cells can vary depending on the donor of the tissue, and rrhRSV can infect the neighbor cells where infected cells secrete cytokines such as IL-6, IL-12, TNF-α, and IFN-γ [42]. Taken together, these in vitro studies support the notion of the potential of hRSV to cross the placenta.

Most of the knowledge about the potential vertical transmission of hRSV comes from animals studies [24,43,44,45]. The hRSV infection of pregnant Fisher 344 rats showed that 30% of the analyzed fetuses have viral RNA in their lungs [24]. The analysis of the viral replication in the fetuses gestated in hRSV-infected rats using the rrhRSV showed the detection of the RFP in 67% of them [24]. However, no lung development changes were observed between the fetuses exposed to hRSV in utero compared to controls [24]. Neurotrophins, such as nerve-growth factor (NGF) and bone-derived neurotrophic factor (BDNF), play an important role not only in the CNS but also in the modulation of inflammation in the lungs [46,47]. Along these lines, about 88% of the fetuses from the rrhRSV-infected rats display a positive correlation between the viral titers and NGF expression, suggesting that the hRSV infection promotes neurotrophic dysregulation [24]. One of the most important findings is that in utero exposure to hRSV promotes airway hyperactivity when reinfected with the virus [24].

All these results show the impact of the hRSV infection on pregnant women and the ability of the virus to cross the placenta, infect the fetus, and induce susceptibility to develop long-lasting lung pathology, including asthma.

## 3. hRSV Infection during Gestation: Possible Impairment of Fetal Neurodevelopment?

According to what was mentioned above, the evidence supports the notion that hRSV can be vertically transmitted and infect the fetus [24,45]. Besides the severe respiratory illness in pregnant women caused by the hRSV infection, a possible consequence of this infection that has not been evaluated is the induction of maternal immune activation (MIA) [48]. Although this is a relevant issue, there is no data about hRSV and MIA relationship. Therefore, this section will discuss the possible association between the infection with hRSV and the development of MIA during pregnancy.

During pregnancy, the viral infection promotes MIA, which has been associated with alterations in the neurodevelopment of the fetus, such as cognitive impairment and neuropsychiatric illness [48,49,50]. Several studies using pregnant animal model exposure to polyriboinosinic-polyribocytidylic acid (Poly: IC) show detrimental effects on the brain of the fetus development, promoting neurological outcomes in adulthood [32,48,51,52,53]. It has been observed that the Poly: IC treatment in animals during pregnancy promote alterations in the neurotransmitters signaling such as dopamine (DA), γ-aminobutyric acid (GABA), and serotonin (5-HT) at different gestational stages [54,55,56,57]. MIA has been related to neuropsychiatric disorders such as schizophrenia (SCZ) and autism spectrum disorder (ASD), where most of the studies have focused [50,58]. Indeed, the offspring of Poly: IC–treated mothers showed an altered performance in behavioral tests, including prepulse inhibition of startle reflex, novel recognition object, and social interaction [55]. Moreover, these offspring showed increased levels of DA in the shell of the nucleus accumbens (Nac), lateral globus pallidus (LGP), the prefrontal cortex (PFC), which can be related to SCZ. [55,59]. Besides, DA receptors (DAR) are decreased in the medial PFC while the tyrosine hydroxylase (TH) expression is increased in the dorsal and ventral striatum [60,61]. Additionally, the effects observed in the offspring of Poly: IC–treated mothers regarding the DA system dysfunctions have a transgenerational impact, showing altered methylation patterns on the TH and nuclear receptor-related 1 protein (*Nurr1*) genes relevant for the DA functions [62]. Regarding the 5-HT signaling in the offspring of Poly: IC–treated mothers, reports showed decreased levels of this neurotransmitter in the frontal cortex and that the expression of 5-HT receptor 2A (5-HT_2A_R) was increased, while no difference was found in the expression of 5-HT_1A_R, 5-HT_1B_R, 5-HT_2B_R and 5-HT_7_R [63]. As described for the DA system, the alteration of 5-HT is related to SCZ. [64]. Furthermore, prenatal exposure to Poly: IC treatment impacts the GABAergic drive onto the hippocampal CA1 pyramidal cells during adulthood and reduces the excitatory synaptic inputs [65].

On the other hand, MIA induced by Poly: IC increases cytokine levels that affect the neurodevelopment of the offspring [66,67]. Cytokines such as IL-1β, IL-10, CCL5, CXCL10, and granulocyte colony-stimulating factor (G-CMSF) increased in the mother serum. Meanwhile, the brain of the fetus presents different cytokines profiles in prenatal related to postnatal; there was an increase in the levels of IL-1β, IL-7, and IL-13 during prenatal while an increase in the levels of IL-2, IL-3, and IL-7 was found in postnatal [66]. The prenatal cytokines profile alteration in the brain of the fetus promotes structural changes in the CNS and impacts the normal cognitive and behavioral process [68]. Thus, although Poly: IC is an excellent model to study MIA and its effect on neurodevelopment and its relationship with neuropsychiatric diseases, this does not respond to a specific trigger of MIA in humans.

Accordingly, it has been described that respiratory viruses, such as the Influenza virus, induce a deleterious effect in both pregnant women and the offspring [32,69,70,71]. Even though the transplacental transmission of the Influenza virus is controversial and it depends on several factors, including the used strain, the fact is that prenatal infection affects the brain development of the fetus [30,70,72,73,74]. This viral agent is one of the respiratory viruses most characterized regarding its effects on the CNS, and it is known to cause neurological symptoms, such as encephalitis, febrile seizures, and delirium [19], so being related to the detrimental development of the brain is not strange. In this context, the viral infection on gestational day 12 (GD12) with the strain Hk-x31 (H3N2) mouse-adapted promotes an exacerbated systemic immune response and disfunction of the major arteries and in the fetus induces a hypoxic state and angiogenesis in the brain [30]. Furthermore, MIA during the Influenza virus infection leads to increased pro-inflammatory cytokines such as IL-6 and IL-17, which have detrimental effects on neurodevelopment [54]. Moreover, it has been reported that MIA can alter the cytokines in maternal serum and the amniotic fluid, placenta, and the fetal brain [75]. Furthermore, results obtained from Influenza virus infection on GD9.5 with H1N1 showed that the adult offspring displayed exploratory and social behavior impairment [32]. Moreover, the use of agonists of metabotropic glutamate receptor 2/3 (mGlu_2/3_R) and 5-HT_2A_R altered the responses to hallucinogens and glutamate antipsychotics, along with an increase of 5-HT_2A_R and a decrease of mGlu_2_R expression in the offspring’s frontal cortex [31]. Like Poly: IC-induced MIA, maternal Influenza virus infection also has been associated with changes in the development of the CNS [50,76], which makes this topic relevant and encourages performing more deep studies in this field.

Previously, we mentioned that hRSV infection could provoke severe respiratory illness in pregnant women. One of the consequences of this infection that has not been evaluated is MIA, which, as described earlier, can trigger an increase of cytokines that can alter the brain development of the fetus. Influenza virus and hRSV can be transplacentally transmitted and associated with neurocognitive and behavioral impairment, whereby it is possible to think that maternal infection with hRSV could affect the brain of the fetus. The lack of studies regarding hRSV infection in this field may be due to the poor search and description of hRSV infection during pregnancy. Therefore, there is no detailed information on how hRSV affects brain development and needs to be evaluated in the future.

## 4. Neurological Consequences due to hRSV Infection

The hRSV-infection not only affects the respiratory tract but can also develop neurological manifestations, which might lead in some cases to long-term neurological sequels [19]. Since the infection with hRSV promotes a more severe disease in infants, the neurological complications that this virus can produce are generally studied in this population [6,19]. In this section, it will be discussed the consequences related to the infection with hRSV on the CNS.

The neurological complications in infants infected by hRSV were firstly reported during a study performed on febrile infants in 1970 [77]. Among these infants, the patients infected with hRSV presented facial palsy and hemiplegia [77]. To date, several reports indicate that the infection with hRSV on infants can lead to the development of ataxia, febrile or epileptic state, encephalitis, and encephalopathies, among other symptoms [19,78,79,80,81]. Among these patients with neurological symptoms, the genetic material of hRSV was found on the cerebrospinal fluid (CSF) and presented elevated cytokines [82,83]. The percentage of acute encephalitis or encephalopathy cases in infants due to the infection with hRSV can reach up to 6.5% [84]. Unfortunately, not all children can recover entirely from the neurological symptoms, and even 7% of the cases result in the death of the patient [84]. Recently, a rare manifestation that contributes to neurological symptoms due to the infection with hRSV was reported [85]. In this report, the two-month-old infant infected by hRSV developed seizures, encephalopathy, and the syndrome of inappropriate antidiuretic hormone secretion (SIADH) [85]. The follow-up performed ten months after the onset of the symptoms showed problems in the patient regarding neuromotor development, mobility, and visual functions [85].

The neuropathologies caused by a viral agent can lead to long-term sequels, such as cognitive and psychiatric disorders, even after the infection has been cleared [86]. In the case of hRSV, patients infected with the virus might develop neuromotor impairment, as previously mentioned, or present learning impairment [85,87]. Among the learning complications reported for hRSV positive cases, a study evaluated the difficulties in learning the native language of infants that suffered from a severe infection within their first six-month-old [87]. During this study, it was observed that patients with severe hRSV-infection under six months of age lead to poor differentiation between native and non-native phonetic, associating the lack of ability to learn the language with memory impairments [87]. Even more, this phenotype lasted up to twelve-month-old and was associated with a detrimental development of communication abilities [87]. A possible explanation for these cognitive consequences comes from a case of an infant infected with hRSV who showed abnormal magnetic resonance imaging (MRI) regarding the hippocampus, which is a zone of the brain that plays a significant role in long-term memory. Its damage contributes to long-term memory impairments [88,89]. Since the ability to learn a language and phonetics is performed through short- and long-term memory [90], it can be suggested that hRSV could have infected the cells located in the hippocampus, which affected the long-term memory, leading to learning language impairment.

Based on what was previously described, it has been suggested that the hippocampus might be involved in developing long-term neurologic consequences due to the infection with hRSV, and in vivo studies were able to demonstrate it [91]. This in vivo study showed that hRSV could reach the brain and be detected in several brain zones, such as the olfactory bulb, cortex, hippocampus, and ventromedial hypothalamic nucleus [91]. The evaluation of the long-term learning impairment was through the Morris water maze (MWM), which demonstrated that 30 days post-infection with hRSV, the rats developed difficulties in spatial learning capacities [91]. It was also shown that the rats infected with hRSV presented a defective long-term potentiation (LTP) response, which is a process that potentiates the long-lasting signaling between the CA_3_ and CA_1_ neurons in the hippocampus [91,92]. This finding provided valuable information regarding the brain region that might be responsible for the long-term memory consequence of the infection with hRSV [91].

Additionally, in vivo studies evaluated the behavioral impairment through the Marble Burying test (MB), which measures the spontaneous and natural behavior of the mice related to the ventral hippocampus function [17,91]. These studies demonstrated that behavioral impairment in the infected mice could be present from 30 to 60 days post-infection with hRSV [17,91]. However, at 90 days post-infection, the mice infected with hRSV showed similar behavior to the control [17,91]. Interestingly, the infection with hRSV modifies the permeability of the blood-brain barrier (BBB) of the mice, which leads to the infiltration of immune cells into the brain and an increased level of pro-inflammatory cytokines, possibly contributing to the behavioral alterations on the mice infected with hRSV [17].

Since it has been established that hRSV is capable of infecting different types of cells from the placenta of pregnant women and cells of the fetus, such as Hofbauer cells [42], it can be suggested that the infection can be passed on to the fetus and cause detrimental development of the CNS from the fetus. These problems in the development of the CNS might make the infants more susceptible to developing neurological symptoms and long-term neurological sequels after infection with hRSV, such as memory and behavioral impairment (Figure 2) [19,87]. Therefore, this option needs to be further evaluated.

## 5. Conclusions

Respiratory infections are a public health problem that causes high rates of hospitalization worldwide every winter season. According to this, hRSV is responsible for ALTRI leading to morbidity and mortality principally in infants and the elderly [1,2]. Besides the knowledge about the hRSV infection in the risk population usually studied, pregnant women have shown that they can suffer a severe respiratory illness caused by hRSV with poorly studied consequences [22,34,93].

Importantly, recent evidence shows that hRSV can infect placental cells, cross the placental barrier, and, therein, be vertically transmitted to the fetus [18,24,40]. One of the unknown relevant issues is the MIA effect on hRSV infection, which is essential since MIA has been associated with cognitive and behavioral impairment and neuropsychiatric illnesses such as SCZ and ASD [32,53,57,94]. The critical information about the harmful effect of MIA on the neurodevelopment of the fetus with neurological consequences in adulthood came from the Poly: IC model, which provides valuable evidence of the alteration of neurotransmitters pathways and cytokines expression that affect brain development [95,96]. However, this model cannot be extrapolated to a specific viral infection in humans. What is known is that the Influenza virus, which infects pregnant women and impacts the fetus principally in the second and third trimester of pregnancy, can trigger MIA in animal models [69,97]. Accordingly, prenatal Influenza virus infections increase maternal cytokines promoting an imbalance in the fetus cytokines that alter the normal neurodevelopment and neurotransmitters signaling, as was observed in SCZ and ASD [31,50].

Regarding the role of MIA in hRSV infection and its impact on the fetus, no studies have been performed. The current knowledge about the hRSV and neurological alterations came from reports showing that patients with hRSV infection have seizures, encephalitis, and encephalopathies [19,98,99]. Besides, elevated cytokines and viral RNA were found in CSF from patients with neurological manifestations [82,83]. In this context, evaluations of the neurological effects of hRSV infection in animal models showed that viral RNA and proteins could be detected in the brain of infected animals [91]. Additionally, the hRSV infection induces an increase in the BBB permeability, allowing the infiltration of immune cells and increased levels of cytokines in the brain of infected animals [17]. Cognitive and behavioral impairments were observed 30 to 60 days post-infection, which could be explained by the possible hRSV effect on infected CNS cells [17]. Due to these results, the first study regarding the possible neurological effects of the hRSV infection was conducted in infants was impairment of language acquisition impairment was found [87]. All these data showed the potential harm that the hRSV infection could be for normal brain development, but there is not enough research in this field.

Importantly, antibodies generated against a virus can be passed towards the offspring, and by doing so, they can confer neurological protection to the fetus [99]. This neuroprotection is achieved since the antibodies can reach the placenta and spread via the umbilical cord to the CNS on the fetus, accumulating these antibodies on the neural tissue [99]. Based on this, the possibility of immunizing women during pregnancy is an area that needs to be evaluated as a possible solution to avoid possible neurological consequences on the infant.

## Figures and Tables

**Figure 1 pathogens-11-00047-f001:**
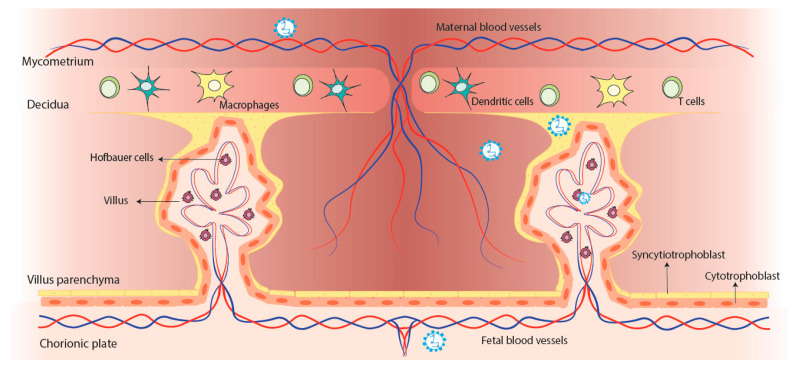
**Potential vertical transmission of hRSV from the mother to the fetus.** The infection with hRSV can travel through the bloodstream until reaching the placenta, where it can infect the Syncytiotrophoblast and the Hofbauer cells, but not the cytotrophoblast. After the infection of the cells from the placenta, it might reach the fetus through the blood vessels of the fetus.

**Figure 2 pathogens-11-00047-f002:**
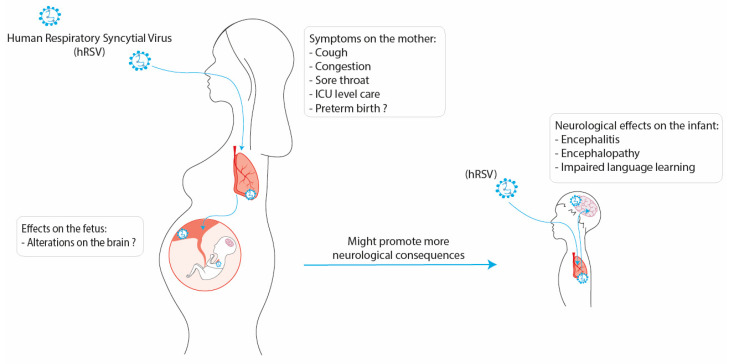
**Possible consequences of the infection with hRSV during pregnancy on the fetus and infant.** The infection with hRSV on a pregnant woman can develop mild symptoms such as cough, congestion, and in the more severe cases, the mother might need ICU-level care. Additionally, the infection with hRSV has been associated with preterm birth. This infection might cause alterations in the development of the fetus and promote more cases of neurological consequences on the infants, which can develop encephalitis, encephalopathy, and impaired language learning.

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
