# Peer review of "Potential Neurocognitive Symptoms Due to Respiratory Syncytial Virus Infection"

_pathogens, 2021, doi:10.3390/pathogens11010047_

Round 1

Reviewer 1 Report

This manuscript overviewed human respiratory syncytial virus (hRSV) infection during pregnancy, gestation and effects on the newborn, especially on the neurological consequences. Overall, the manuscript has some scientific contribution. However, revisions are needed before acceptance.

Major revision:

  1. More detail descriptions on how hRSV affect the brain development and cause the neurocognitive symptoms are needed.
  2. It would be great if the authors can compare the hRSV to other pathogens which can cause neurocognitive symptoms on etiology and pathogenesis.  

Author Response

Answer to Reviewer 1

1.- Reviewer 1: More detailed descriptions on how hRSV affect brain development and causes the neurocognitive symptoms are needed.

Answer: As was requested by the Reviewer, we clarified that there is no detailed description regarding how hRSV affects brain development (Page 6, Lines 207-208).

2.- Reviewer 1: It would be great if the authors can compare the hRSV to other pathogens which can cause neurocognitive symptoms on etiology and pathogenesis.  

Answer: As requested by the Reviewer, the etiology and pathogenesis of another virus was included to compare them with hRSV (Page 2 and 5, Lines 69-71 and 182-185, respectively).

We would like to thank the Reviewers and the Editors for their time and effort in handling this manuscript and hope that the current revised manuscript is acceptable for publication in Pathogens.

Reviewer 2 Report

The authors discussed hRSV infection during pregnancy and suggested that it may cause neurocognitive symptoms.

Major:

  • 131-133: "All these results show the impact of the hRSV infection on pregnant women and the ability of the virus to cross the placenta, infect the fetus, and induce susceptibility to developing long-lasting lung pathology, including asthma."
    • The authors must cite evidence that hRSV causes asthma susceptibility.
  • Because there is no evidence to support hRSV-induced MIA, the entire section 3 reviewing the effect of MIA in hRSV-induced neurological changes is overly reliant on speculation.
    • Please consider moving the paragraph 193-201 to the beginning of section 3, so that readers who are not interested in MIA can skip it.

Author Response

Answer to Reviewer 2

1.- Reviewer 2: 131-133: "All these results show the impact of the hRSV infection on pregnant women and the ability of the virus to cross the placenta, infect the fetus, and induce susceptibility to developing long-lasting lung pathology, including asthma." The authors must cite evidence that hRSV causes asthma susceptibility.

Answer: As requested by the Reviewer, we modified the manuscript to include evidence regarding the susceptibility of asthma caused by hRSV (Page 2, Lines 81-82).

2.- Reviewer 2: Because there is no evidence to support hRSV-induced MIA, the entire section 3 reviewing the effect of MIA in hRSV-induced neurological changes is overly reliant on speculation.

Answer: As was requested by the Reviewer, we have modified the manuscript to clarify that in this section, we will discuss the possible association between MIA and hRSV (Page 4, Lines 136-143).

3.- Reviewer 2: Please consider moving paragraphs 193-201 to the beginning of section 3 so that readers who are not interested in MIA can skip it.

Answer: As requested by the Reviewer, the paragraph was moved to the beginning of section 3 (Page 4, Lines 136-141).

We would like to thank the Reviewers and the Editors for their time and effort in handling this manuscript and hope that the current revised manuscript is acceptable for publication in Pathogens

Round 2

Reviewer 1 Report

The revised version of the manuscript appears to be good.